# *Corynebacterium striatum* Periprosthetic Hip Joint Infection: An Uncommon Pathogen of Concern?

**DOI:** 10.3390/healthcare12020273

**Published:** 2024-01-21

**Authors:** Athanasios Galanis, Spyridon Karampitianis, John Vlamis, Panagiotis Karampinas, Michail Vavourakis, Christos Vlachos, Eftychios Papagrigorakis, Dimitrios Zachariou, Evangelos Sakellariou, Iordanis Varsamos, Christos Patilas, Sofia Tsiplakou, Vasiliki Papaioannou, Spyridon Kamariotis

**Affiliations:** 13rd Department of Orthopaedic Surgery, National & Kapodistrian University of Athens, KAT General Hospital, Kifisia, 14561 Athens, Greece; athanasiosgalanis@yahoo.com (A.G.); spyroskarampitianis@gmail.com (S.K.); jvlamis@email.com (J.V.); karapana@yahoo.com (P.K.); christosorto@gmail.com (C.V.); efpapagr@hotmail.com (E.P.); dimitriszaxariou@yahoo.com (D.Z.); vagossak@hotmail.com (E.S.); jordan.var1995@gmail.com (I.V.); chris_pat7@hotmail.com (C.P.); 2Department of Microbiology, KAT General Hospital, Kifisia, 14561 Athens, Greece; stsiplakou@gmail.com (S.T.); bettypap@otenet.gr (V.P.); spyroskam@hotmail.com (S.K.)

**Keywords:** *Corynebacterium striatum*, prosthetic joint infection, hip, sonication, immunodeficiency, hip infection, *Corynebacterium*, rare pathogen

## Abstract

Background: Total hip arthroplasty is indubitably a dominant elective surgery in orthopaedics, contributing to prodigious improvement in the quality of life of patients with osteoarthritis. One of the most potentially devastating complications of this operation is periprosthetic joint infection. Immunocompromised patients might be afflicted by infrequent low-virulence organisms not typically detected with conventional procedures. Consequently, employing advanced identification methods, such as the circumstantial sonication of orthopaedic implants, could be crucial to managing such cases. Case Presentation: We present a peculiar case of a 72-year-old female patient suffering from a chronic periprosthetic hip infection due to *Corynebacterium striatum*. The pathogen was only identified after rigorous sonication of the extracted implants. The overall management of this case was immensely exacting, primarily because of the patient’s impaired immune system, and was finally treated with two-stage revision in our Institution. Literature Review: Although copious literature exists concerning managing periprosthetic hip infections, no concrete guidelines are available for such infections in multimorbid or immunocompromised patients with rare low-virulence microorganisms. Hence, a diagnostic work-up, antibiotic treatment and appropriate revision timeline must be determined. Sonication of extracted implants could be a powerful tool in the diagnostic arsenal, as it can aid in identifying rare microbes, such as *Corynebacterium* spp. Pertinent antibiotic treatment based on antibiogram analysis and apposite final revision-surgery timing are the pillars for effective therapy of such infections. Clinical Relevance: *Corynebacterium striatum* has been increasingly recognized as an emerging cause of periprosthetic hip infection in the last decade. A conspicuous rise in such reports has been observed in multimorbid or immunocompromised patients after the COVID-19 pandemic. This case is the first report of *Corynebacterium striatum* periprosthetic hip infection diagnosed solely after the sonication of extracted implants. This paper aims to increase awareness surrounding *Corynebacterium* spp. prosthetic joint infections, while highlighting the fields for further apposite research.

## 1. Introduction

Total hip arthroplasty (THA) is regarded as one of the most common elective operations performed worldwide. Specifically, linear progression models indicate a 71.2% upsurge in THA volume by 2030 in the United States alone [1]. Although THA unequivocally yields significant outcomes in ameliorating the quality of life in patients with osteoarthritis (OA) and has been characterized as the “operation of the century”, this procedure is not without its complications [2,3]. The most ordinary complications after THA include wound complications, thromboembolic disease, dislocation, periprosthetic fracture(s), periprosthetic joint infection and implant loosening [4].

A periprosthetic joint infection (PJI) is an infection involving the prosthetic joint or the surrounding tissues. Notwithstanding that PJI is not confoundedly accustomed, it is considered one of the most dreaded complications following THA due to the correlated high morbidity, mortality and upraised economic burden, as the requirement for reoperation(s) is exceedingly escalated [4,5]. Obesity, poor preoperative glycemic control, alcohol use, a history of previous infections, improper sterilization procedures and negligent surgical techniques have been described as the main reasons behind PJIs, while the Staphylococcus aureus complex and coagulase-negative Staphylococcus species have been reported as the most dominant pathogens causing PJIs [4,5,6]. Accurate PJI diagnosis after total joint arthroplasty has been a significant challenge in orthopaedic surgery due to the alterability in clinical presentation and the profusion of available diagnostic pathways and tests [4,6]. Despite the established progress in the diagnostic methods for the most common Gram-positive bacteria involved in PJI, such as Staphylococcus aureus and Staphylococcus epidermidis, existing literature on atypical pathogens needs to be revised [4,5,6].

Corynebacterium species are facultative anaerobic Gram-positive bacilli located ubiquitously on human skin and mucous membranes. It is of the utmost significance that many laboratories do not routinely investigate and identify Corynebacterium species, as they are typically considered contaminants when isolated in cultures [7]. This policy, limited by physicians’ suspicion and in many cases the lack of contemporary laboratory equipment and techniques, can justify the extremely low number of PJIs that have been attributed to Corynebacterium species in the existing literature [7,8,9,10]. Nonetheless, there is augmented interest regarding Corynebacterium-associated PJIs, which are increasingly recognised as an atypical cause of PJI in orthopaedic surgery [8,9,10]. Taking into account the fact that Corynebacterium species are part of normal skin microbiota, the arduousness of proper cultivation and the absence of a gold-standard diagnostic test, the knowledge surrounding Corynebacterium-related PJIs after THA is broadly narrow [11]. However, it has been delineated that chronically ill or immunocompromised patients can be affected in higher rates by various infections caused by *Corynebacterium* spp. (including striatum), such as endocarditis, pneumonias and prosthetic joint infections [8,9,10,11].

We present a markedly rare case of a *Corynebacterium striatum* PJI after THA which was treated at our Institution, along with a brief review of the relevant literature available. Diagnosis of a PJI was attained using the sonication method with a low-frequency ultrasound (35–40 kHz) applied to the removed implants, followed by sonication fluid cultures. This is the first report of post-THA PJI with *Corynebacterium striatum* diagnosed only in sonication fluid cultures of the extracted implants. Presenting this thought-provoking case, this paper aims to enhance comprehension and awareness of Corynebacterium-associated PJI and accentuate the necessity for further pertinent research.

## 2. Case Presentation

A 72-year-old female patient presented at the Outpatient Clinic of our Department due to a suspected wound infection and extensive groin pain four years after undergoing an elective right THA for hip osteoarthritis at another institution. The patient`s past medical history was notably considerable and involved hypertension, obesity (BMI > 36), a gastric ulcer, lung fibrosis, rheumatoid arthritis and hospitalization for pneumonia which was treated with intravenous (IV) antibiotics. More specifically, the prescription medications for the rheumatoid arthritis included leflunomide and rituximab, whilst lung fibrosis was managed with nintedanib.

Regarding her orthopaedic medical history, four years previously, the patient underwent a right THA at a private institution through a standard Anterior Minimally Invasive Surgery (AMIS) approach (Figure 1). From the first postoperative days, serous wound drainage was discerned. Following that, she was reoperated on twice at the same hospital and by the same team, thirty days and forty-five days after the initial surgery, where wound exploration, debridement and washout with implant retention were performed due to residual pain and incessant wound leakage. Subsequently, the AMIS wound was not entirely healed, and the patient continued to complain about wound drainage and hip pain. She was discharged home with vacuum-assisted wound closure (VAC) and per os antibiotics. Finally, successful wound healing was achieved roughly four months after the primary THA.

The patient presented at our clinic complaining about persistent right-hip pain for the last three years, intermittent serous fluid leakage from specific parts of the AMIS wound and baffling swelling. She has not been able to perform full weight-bearing since. Upon clinical examination, a limited range of hip motion was revealed (flexion 50 degrees, extension 10 degrees), and a sinus tract formation on the distal part of the wound was described. She was promptly admitted to our department, and a swift decision was made for absolute metalwork removal and exchange with a gentamicin-preloaded spacer after confirming that the patient had not received any antibiotics in the last six months. Concerning the lab workup, the patient`s white blood cell count (WBCs) and C-reactive protein (CRP) on admission were 11.6 × 103/μL and 7.2 mg/dL, respectively. Given that the patient might need more hip operations and a revision THA was requisite should the infection be eradicated, a conventional posterior hip approach was utilised for better and greater exposure of the hip joint and adjacent soft tissues. After punctilious washout, bone and soft tissue debridement, and complete metalwork extraction, nine (9) standard bone and soft tissue cultures were sent for analysis and culture. At the same time, a hip spacer (Spacer G, Tecres SpA, Verona, Italy) was implanted as planned (Figure 2).

Concerning the whole sonication procedure, the explanted components of the prosthesis were aseptically removed in the operating room, stored in a sterile, airtight container and then transported to our Institution’s highly developed laboratory. The extracted metalwork included the femoral stem, head, acetabular shell and polyethylene. Sonication of the implants was executed using the procedure presented by Trampuz et al. [12]. Sterile Ringer solution was added to the container in a laminar airflow biosafety cabinet to cover 85–90% of the implants’ volume. The container was vortexed for 30 s and then subjected to sonication for 1 min (frequency, 40 kHz; and power density, 0.22 W/cm^2^, as defined with the utilisation of a calibrated hydrophone [type 8103, Brüel and Kjær, Naerum, Denmark]). The BactoSonic ultrasound bath (Bandelin GmbH, Berlin, Germany) was employed for sonication in compliance with the manufacturer’s instructions (http://www.bactosonic.info/ (accessed on 27 November 2023)). The container was vortexed for 30 s to remove residual microorganisms and for homogeneous distribution in the sonication fluid. A total of 0.1 mL of sonication fluid was inoculated on blood agar plates, which were then incubated at 37 C aerobically, anaerobically and at 5% Co_2_ conditions. Moreover, 1 mL of the remaining sonication fluid was added to 10 mL of thioglycolate broth (TGB). Microorganisms were enumerated and classified with routine microbiologic techniques.

Forthwith, postoperatively, initial empirical IV therapy with meropenem and teicoplanin were commenced. WBCs and CRP on the third postoperative day were 10.84 × 103/μL and 10.5 mg/dL, respectively. The inflammatory markers were decreasing steadily, and on the 13th postoperative day, WBCs and CRP were 7.20 × 103/μL and 4.1 mg/dL, respectively. The postoperative course was uneventful, and the empirical antibiotic treatment was continued until the lab outcomes we complete. To our salient surprise, six days postoperatively, all nine bone and soft tissue cultures resulted in negative. Being confident about PJI, we kindly asked the lab whether it could analyse these samples again. However, there was no difference after they rechecked the cultures twice, waiting for the sonication outcomes.

Contrariwise, 15 days postoperatively, sonication of the implants denoted growth of small, white, and smooth colonies (>100 cfu/mL), which were Gram-positive bacilli with typical coryneform morphology. Identification was carried out employing the API Coryne (bioMérieux, Marcy l’Etoile, France), and MALDI-TOF MS (Microflex LT, Bruker Daltonics, Bremen, Germany Bruker Biotypes, Germany) systems, and the isolate was recognized as *Corynebacterium striatum*. It was a strikingly unanticipated result and PJI pathogen.

Susceptibility testing was conducted using gradient MIC test strips (Liofilchem, Italy) for the following antibiotics: benzylpenicillin, ciprofloxacin, clindamycin, daptomycin, linezolid, moxifloxacin, rifampicin and vancomycin, and by the disk diffusion method for ciprofloxacin, clindamycin and rifampicin. The isolate demonstrated susceptibility to linezolid, vancomycin and daptomycin but was resistant to benzylpenicillin, ciprofloxacin, clindamycin, rifampicin, moxifloxacin and tetracycline (EUCAST Clinical Breakpoints 2023). The minimum inhibitory concentration (MIC) value for daptomycin was 0.125 μg/mL (Figure 3).

Following this resounding result and the expeditious consultation of our Department’s infectious disease specialist, the preceding antibiotic treatment was halted, and monotherapy with IV daptomycin was continued for four (4) more weeks. After an uncomplicated postoperative period, the patient was discharged from our Department 6 weeks postoperatively without any signs of wound infection or declining inflammatory markers. It was arranged for the patient carry on IV daptomycin at home for another six weeks and have several physio sessions. Consistent antibiotic treatment, for a total of 3 months, was completed as instructed. At the 3-month follow-up visit, the patient presented with satisfactory wound healing with no signs of inflammation and low inflammatory markers, along with remaining groin pain and a comparatively confined range of motion owing to the presence of the spacer.

At that point, the patient was explicitly advised about the requirement for further successive follow-ups to evaluate wound condition, inflammatory marker values after antibiotics discontinuation and general clinical condition. The plan was to perform a revision THA surgery in a few months only after it was assured that all inflammatory markers were consistently negative and no wound or clinical complications were observed. Hence, it would be firmly decided that the hip infection was terminated, and it was safe to proceed to the final treatment stage with a revision THA.

However, four months postoperatively, the patient was admitted to another institution with a severe respiratory infection. Blood cultures were positive for both *Staphylococcus* spp. and *Streptococcus* spp. She remained hospitalized, receiving IV antibiotics for more than a month. As a result, proper evaluation of inflammatory markers was not feasible. The patient was then hospitalized again with a urinary tract infection after a month, exhibiting positive urine cultures for *Pseudomonas* spp. A precise orthopaedic follow-up after the first four months postoperatively was not obtainable since the patient had numerous hospitalizations during the first postoperative year, suffering from infections in different parts of the body, including respiratory, urinary and gastrointestinal systems, with other pathogens involved. It is pivotal to highlight that no *Corynebacterium striatum* or *Corynebacterium* spp. infection was revealed during her hospitalizations and scrupulous testing.

Taking into deliberation the presence of recurrent copious infections in our patient during the first postoperative year, the presence of an atypical pathogen responsible for the PJI and her entire past medical history, it was deduced that the patient might feature an impaired immune system. Consequently, the risk of revision THA surgery was regarded as highly elevated. She was referred to a preeminent immune system specialist physician. Still, no solid inferences could be drawn to explain the multiple infection incidents, albeit without excluding the notion of a weakened immune system. The patient continued to complain about considerable hip pain and an inability to sustain full weight bearing since she had the hip spacer for an extended period and insisted on having a new hip operation. She was thoroughly counselled that, if a revision THA was performed, the peril of a potential new PJI was exceptionally high. Eighteen months from the spacer placement, the patient featured no clinical infections or hospitalizations during the last 5.5 months with negative inflammatory markers and no indications of hip wound infection. After diligent consent, the patient was reoperated on, and a typical revision THA (Avenir Femoral Stem and ZCA All-Poly Acetabular Cup, Zimmer Biomet Holdings Inc., Warsaw, Indiana, United States) was executed utilizing the same posterior approach that was employed in the first operation at our Institution (Figure 4). Intraoperatively, no evident indications of remaining infection were observed regarding the bones and surrounding soft tissues, while bone stock was sufficient. In case that any tissue or fluid with the suspicion of infection was presented in the operating field, no new arthroplasty materials would have been implanted.

Multiple bone and soft tissue cultures were taken during the operation, and the spacer was sent for sonication. The postoperative course was uneventful, and both cultures and the sonication returned negative, while the patient was administered IV daptomycin for one month postoperatively. Finally, the patient’s 1-year follow-up revealed no signs of PJI and a sufficiently good range of hip motion, with the patient being capable of unhindered full weight-bearing and very satisfied with her hip functional capacity.

## 3. Discussion

When isolated from clinical materials, non-diphtheria Corynebacterium species are preponderantly regarded as contaminants with a contentious potential to trigger infection [13,14]. The small-scale incidence of corroborated PJIs ascribed to non-diphtheria Corynebacterium species, the substantial diagnostic barriers and the lack of standardized guidelines pose challenges regarding the fruitful management and treatment of such infections [11,15]. Nonetheless, as data from the literature expands and microbiological lab procedures evolve, they are being recognized as an emerging pathogen to blame for musculoskeletal infections and PJIs [16]. Published literature concerning confirmed PJIs with Corynebacterium species is limited to the last 15 years, whilst over 70% of these papers have been published in the previous three years [8,9,10,17]. This report demonstrates the diagnostic process, management and pitfalls of an infected THA with a distinctly rare low-virulence microbe, such as *Corynebacterium striatum*. To the best of our knowledge, this is the first reported periprosthetic hip infection with *Corynebacterium striatum* diagnosed solely with a sonication fluid culture.

In general, PJI diagnosis requires the fulfilment of explicit criteria. In 2018, Parvizi et al. described a novel definition of hip and knee PJIs grounded in evidence-based and validated criteria [18]. The presence of a sinus tract or two (2) positive cultures of the same microorganism is considered a major criterion and, therefore, indicative of PJI. Our patient presented to our Outpatient Clinic after noticing obtrusive fluid leakage from the wound of her primary THA. A plain clinical examination revealed the existence of a small sinus tract. Consequently, the patient was immediately admitted to our Department after a high suspicion of PJI. According to the PJI definition, serum and synovial fluid analysis provide the minor criteria for diagnosing PJI. In particular, serum levels of CRP >1 mg/dL, D-dimer >860 ng/mL, erythrocyte sedimentation rate >30 mm/h, or synovial fluid WBCs >3000 cells/μL, alpha-defensin signal-to-cutoff ratio >1, leukocyte esterase ++, polymorphonuclear percentage >80% and synovial CRP >6.9 mg/L account for the minor criteria. In our case, the diagnosis was made straightforwardly and clinically. At the same time, inflammation markers, such as CRP and WBCs, were used to assess response to treatment only during the first four months postoperatively when the patient was not suffering from the succeeding various infections in other systems, which rendered the evaluation of inflammation markers impossible.

It was suggested that our patient’s immune system might be impaired, which could be attributed to long-term treatment with leflunomide, rituximab and nintedanib after she was diagnosed with rheumatoid arthritis and lung fibrosis. This fact could explain the unexpected infrequent pathogens detected in the sonication, as well as the various infections in different systems of our patient’s body after the spacer implantation, which incommoded our treatment strategy and overall management of the case. Apart from that, it has been reported that patients with inflammatory arthritis, such as rheumatoid arthritis, feature a higher risk of PJI after orthopaedic surgery [19,20]. PJI in immunocompromised or multimorbid patients is a complex scenario for orthopaedic surgeons since limited reliable data are available on appropriate treatment algorithms, and vast variability has been observed among the reported cases [19]. Thus, our present-day knowledge is based merely on published reports with pervasive heterogeneity. In our case, we opted for complete metalwork removal and implantation of a gentamicin-loaded spacer. It is essential to underline that previous treatment with debridement, antibiotics and implant retention (DAIR) had already been employed after the primary THA with poor outcomes. Orthopaedic surgeons should be highly vigilant when implementing DAIR in immunocompromised or multimorbid patients, as there are no widely established guidelines for treating PJI in such groups [20]. On the one hand, the long-term antibiotic treatment required for the eradication of *Corynebacterium striatum*-associated PJI in immunocompromised patients is decidedly demanding owing to the potential side effects of the treatment and the consequent necessity to modify the antibiotic regimens in the already-narrowed antibiotic arsenal [21]. On the other hand, a diligent assessment of all risk factors is vital before an invasive approach is eventually decided. A revision THA might induce debilitating consequences in immunocompromised or multimorbid patients, as it has already been correlated to increased infection rates compared to revision THA in immunocompetent individuals [22]. In our case we were reluctant to proceed with revision surgery due to the complicated nature of the case; however, the revision operation proved productive, and the patient was delighted with the outcome.

Treatment options for hip PJI are confined to DAIR, 1-stage revision and 2-stage edit [23]. Current data in terms of the appropriate timeline for DAIR are inconclusive. On the one hand, duration of symptoms >4 weeks and late-onset PJI are regarded as chief risk factors for DAIR failure [24,25]. On the other hand, published literature connotes that DAIR could be an applicable option for infection control, regardless of the time frame [23]. In our case, we decided to implement a 2-stage revision for the PJI treatment after gingerly examining all treatment alternatives. Our patient exhibited an early chronic infection, while DAIR treatment had already been performed in another institution after the primary THA. The other option was a single-stage revision (SSR). Although various reports suggest that SSR for PJI after THA features similar infection rates compared to 2-stage revision with minor resource utilization [26,27], it should be limited to cases where the organism responsible for the PJI is perspicuously recognised and identified with detailed sensitivities and bacterial MICs [28]. In our case, a difficulty that we encountered was that the pathogen was unspecified until the 15th postoperative day, when the sonication cultures identified *Corynebacterium striatum*, and we had no microbiological data from the patient’s previous operations and hospitalizations. In addition, all bone and soft tissue cultures were negative, whilst until the sonication findings only administration of empirical antibiotic treatment was feasible. On the other hand, no intraoperative surgical technical difficulties were coped regarding both surgeries executed in our Institution, as the implants from the primary THA were extracted normally and the condition of soft tissues and bone stock in the revision surgery was satisfactory.

Sonication of the implants could be a potent tool in the diagnostic procedure for PJIs in immunocompromised or multimorbid patients. In our case, it was tremendously surprising that all soft tissue and bone cultures demonstrated no growth of microorganisms. Although the gold standard regarding pathogen identification in PJI is microbiological cultures, failure to distinguish the microbe(s) occurs in 5 to 45% of cases [29,30,31,32]. Factors contributing to false-negative culture results involve (1) previous antibiotic administration, (2) microorganisms that produce biofilms, (3) insufficient culture means for atypical organisms, and (4) improper culture handling or sample transfer to laboratory [30,32]. Immunocompromised and multimorbid patients are prone to community and opportunistic infections, including PJIs, from low-virulence atypical microbes that require hospital admission and apposite antibiotic treatment. These atypical microorganisms are widely characterised by an increased risk for negative tissue culture results [30,31,32]. According to Trampuz et al., the culture of samples obtained from sonication of prostheses is more sensitive than conventional bone and soft tissue cultures, principally in patients treated with antibiotics before surgery [12]. Sonication is a precious method that can disrupt biofilms on an explanted prosthesis, meliorating the yield of culture. Contemporary published literature has denoted that sonication is exceedingly valuable for detecting persistent post-revision occult infections and in terms of guidance for administering the proper antibiotic regimen [33]. In our case, despite all soft tissue and bone cultures being negative, sonication cultures of the extracted implants successfully identified the growth of *Corynebacterium striatum*, aiding us in creating an effective treatment plan. Corynebacterium species are generally resistant to many antibiotics. *Corynebacterium striatum* is commonly reported as being multidrug-resistant, with a phenotype which is resistant to penicillins, carbapenems, aminoglycosides, fluoroquinolones, lincosamides and macrolides but susceptible to glycopeptides, tigecycline, daptomycin and linezolid [13,14,15,16]. This is due to the antibiotic susceptibility pattern of the isolates from our case.

*Corynebacterium striatum* is a rare yet severe cause of PJI, predominantly seen in immunocompromised and multimorbid patients. It has been suggested that patients that are considered chronically ill or who have an impaired immune system from immunosuppressive medication feature a higher incidence of *Corynebacterium striatum*-associated copious infections (more commonly respiratory tract infections), including PJIs [13,15,16]. It is typically considered a contaminant, so many laboratories do not routinely identify or search for *Corynebacterium* spp. Sonication of the implants could be a powerful tool in managing culture-negative PJIs. Precise identification of pathogens and analysis of antibiotic resistance are considered of paramount significance as they steer antibiotic selection [29,30,31,32,33]. Laboratory staff should be highly diligent and supplied with advanced equipment to discern PJIs from infrequent atypical microbes, such as *Corynebacterium striatum*, as their incidence is probably not as rare as presented in the current literature, being underdiagnosed. In our case, the laboratory techniques employed in our Institution, such as the contemporary sonication fluid culture procedures and the utilization of API Coryne and MALDI-TOF MS systems, proved efficient in identifying the atypical pathogen responsible for the infection. Of note, more than 70% of the literature concerning *Corynebacterium* spp. PJIs has been published over the last three years. This might indicate an increased incidence of PJIs with low-virulence pathogens after the Coronavirus-19 (COVID-19) outbreak, notably in immunocompromised patients. More specifically, a recent epidemiological study by Orosz et al. [34] demonstrated a significant rise in *Corynebacterium striatum* infections during the COVID-19 pandemic. In terms of future research directions, this could serve as fertile soil for further research to distinguish potential correlations between COVID-19 infection and the late considerable upsurge of *Corynebacterium striatum*-related infections, including PJIs. It is also crucial to investigate whether this recent rise could be justified simply by the latest enhancements in laboratory equipment and augmented physicians’ suspicions of *Corynebacterium* spp. infections. Therefore, a lot of future research in this direction is needed to elucidate the ambiguity in these areas. Under these circumstances, *Corynebacterium striatum* could be considered an emerging pathogen responsible for PJIs in orthopaedic surgery. Finally, it is of utmost importance to emphasize that any PJI provoked by *Corynebacterium striatum* could indicate a patient’s impaired immune system. Hence, orthopaedic surgeons should be very diligent in terms of managing these cases and determining whether standardized treatment protocols should be applied.

## 4. Conclusions

*Corynebacterium striatum* is a low-virulence microorganism that is ordinarily recognized as a contaminant when isolated in cultures. Notwithstanding, it has been lately identified as an emerging cause of PJIs, chiefly in immunocompromised or multimorbid patients. Sonication of implants could be a robust tool for apposite diagnosis of *Corynebacterium striatum* PJI, predominately when soft tissue and bone cultures are negative. Owing to the rarity of such cases, published pertinent literature is exceedingly narrow and features high heterogeneity. Hence, there are no determined guidelines regarding the treatment of uncommon PJIs, specifically in immunocompromised patients. Nonetheless, two-stage revision and intravenous antibiotic treatment remain broadly regarded the mainstay of treatment in cases of late or extraordinary PJIs. Orthopaedic surgeons should be particularly vigilant in considering rare low-virulence organisms in culture-negative PJIs, especially in patients with known or suspected impaired immune systems. It is crucially important that an abrupt upsurge of *Corynebacterium striatum* infections during the COVID-19 era has been observed, triggering the requirement for further investigations. More research needs to be conducted to identify the optimal time frame for revision surgery in immunocompromised and multimorbid patients presented with chronic PJI.

## Figures and Tables

**Figure 1 healthcare-12-00273-f001:**
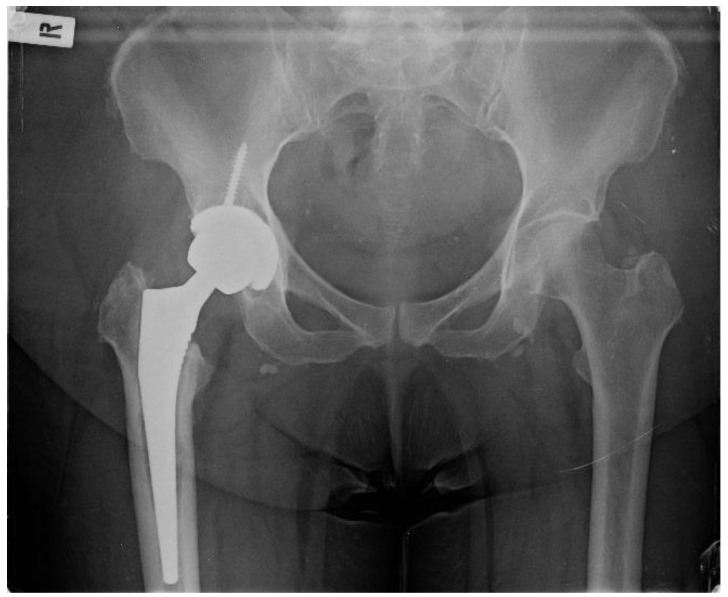
Pelvis X-ray on admission date.

**Figure 2 healthcare-12-00273-f002:**
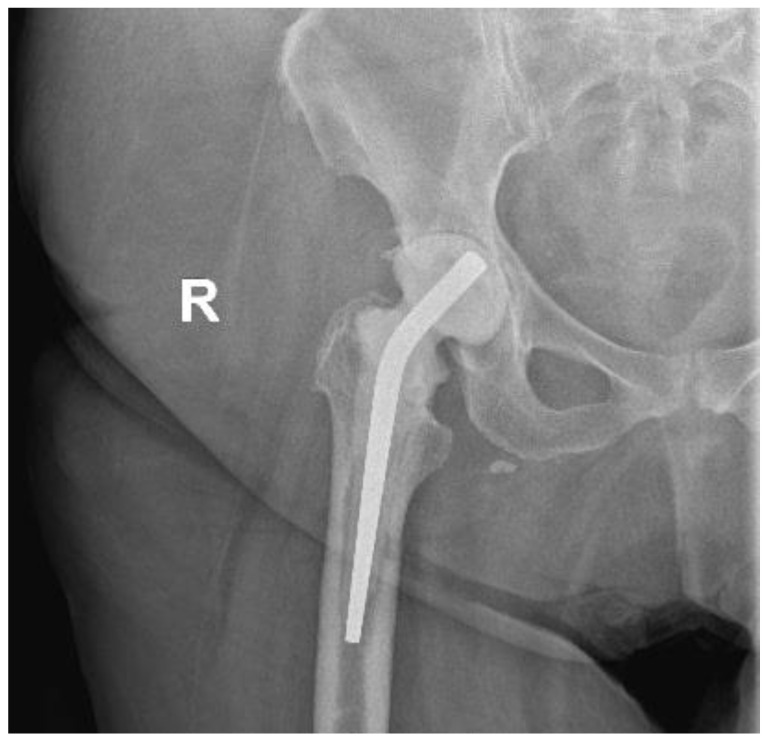
Hip X-ray after arthroplasty removal and implantation of the spacer.

**Figure 3 healthcare-12-00273-f003:**
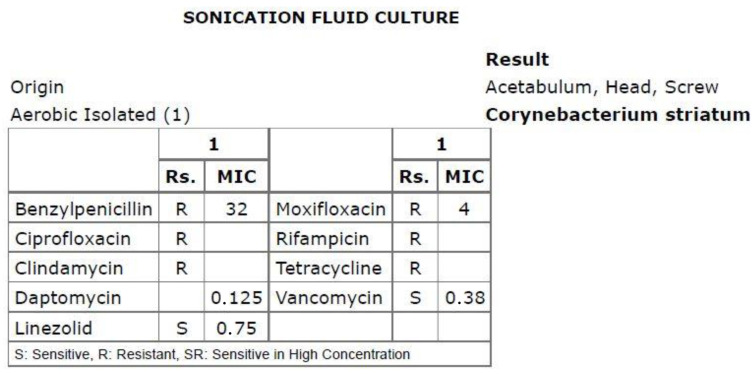
Antibiogram based on the sonication fluid culture.

**Figure 4 healthcare-12-00273-f004:**
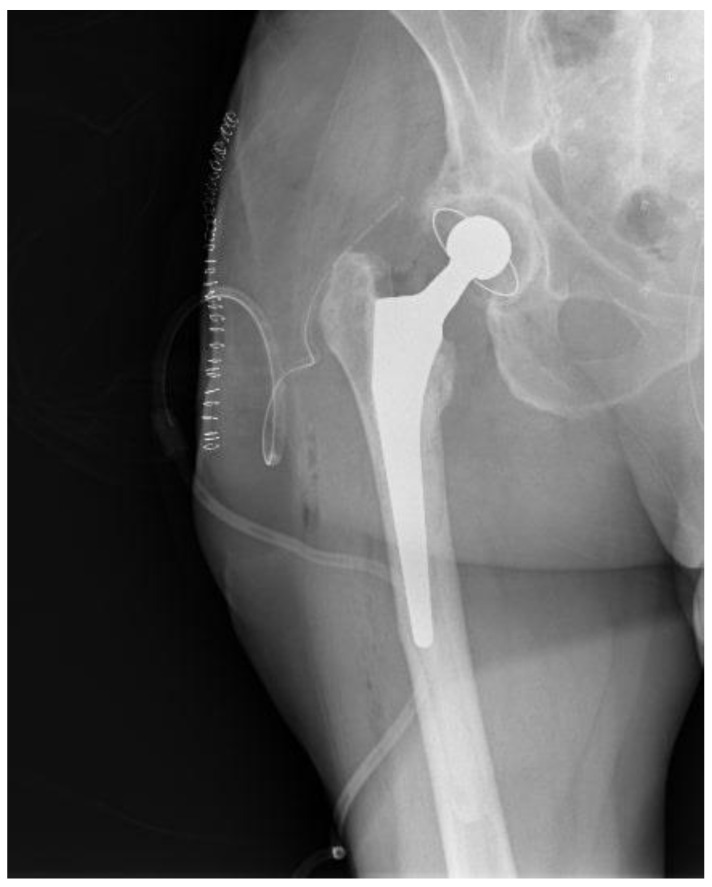
Postoperative hip X-ray after revision THA.

## Data Availability

All raw data are available to access should they be requested.

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
