# Peer review of "Corynebacterium striatum Periprosthetic Hip Joint Infection: An Uncommon Pathogen of Concern?"

_healthcare, 2024, doi:10.3390/healthcare12020273_

Round 1
Reviewer 1 Report
Comments and Suggestions for Authors
Dear authors,
The abstract is well written.
The introduction transposes the research into the topic and formulates the objective of the study at the end.
The case report is clearly presented. However, Could you add information about the spacer and hip prosthesis that was used: type and company. Also, could you provide information about surgical procedure when hip arthroplasty was performed?
The discussions interpret the cases and relate them to other results from the scientific literature. Limitations of the study are presented at the end of the section. A comparison with newly articles in the field is made.
Also, future research directions on this topic could be also stated.
The conclusions are concise and clear.
Author Response
Dear Reviewer 1,
We sincerely appreciate your comments and expertise. thank you very much for the time you spent in evaluating our paper and for your valuable feedback, aiding us in improving the quality of our article. we revised our paper according to your comments.
As instructed, we provided specific information about spacer and hip prosthesis concerning the operations conducted in our institution. However, we do not have these information about the first surgery of the patient (the aims primary tha), as it was carried out in another institution (private). Also, we provided more information about the rtha finally performed. finally, we stated more clearly the future research directions in the discussion section as you indicated.
Thank you very much again for all your assistance.
Yous Sincerely,
The Authors
Reviewer 2 Report
Comments and Suggestions for Authors
In the present MS, the authors describe a 72-year-old female patient with a chronic periprosthetic hip infection due to Corynebacterium striatum and review the pertinent literature. In addition, the authors highlight the role
of Sonication of extracted implants for the identification of rare microbes, such as Corynebacterium spp.
Introduction: please explain PCR.
The case is very well described.
Can the section on Case Report (be shortened? For example, as sonication of the implants was made as described previously (ref. 12), for this reviewer lines from 120 to 131 can be shortened.
Can a table from another article published as such? However, for this reviewer Table 1 is not needed.
Comments on the Quality of English LanguageMinor editing of English language required.
Author Response
Dear Reviewer 2,
We sincerely appreciate your comments and expertise. Thank you very much for the time you spent in evaluating our paper and for your valuable feedback, aiding us in improving the quality of our article. we revised our paper according to your comments.
In the introduction section, we removed the word pcr and rephrased the sentence appropriately as instructed. Also, we chose not to remove lines 120-131 about the details of the sonication fluid culture, as the other reviewers have insisted that these details are essential in our paper. Finally, we removed table 1 completely as you indicated.
Thank you very much again for all your assistance.
Yours Sincerely,
The Authors
Reviewer 3 Report
Comments and Suggestions for Authors
Comments on the Quality of English LanguageAuthor Response
Dear Reviewer 3,
We sincerely appreciate your comments and expertise. Thank you very much for the time you spent in evaluating our paper and for your valuable feedback, aiding us in improving the quality of our article. We revised our paper according to your comments.
As instructed, in the abstract section we shortened its length and improved the clarity of the information.
In the introduction, we stated clearly the higher incidence of corynebacterium striatum pjis in immunocompromised patients and explained the reasons why these infections are reported rare in the existing literature. Alos, literature information on potential sources of infection and microorganisms was included. Finally, we state the goals of our paper clearly accentuating that our case is only a thought-provoking case, and we have underlined the need for further research in specific areas throughout the manuscript and more specifically in the final paragraph of the discussion section.
In the case presentation section, it is stated clearly that the identification of the pathogen was carried out utilizing the api coryne and maldi-tof ms systems, hence, a rna/dna analysis was conducted and the diagnosis was not morphologic. more research is requisite as highlighted thgoughout the paper. in the discussion section, the implications of our patient suffering from ra have been analyzed.
Thank you about your comments in terms of clinical and radiological data collected. the role of ra as a risk factor for pji has been discussed more extensively as you indicated.
In the discussion section, we discussed the limitations of our case reports to a greater extent as instructed. the laboratory technique employed for the identification of the microbe has been described in detail in the case presentation section and analyzed in the discussion section. also, we underlined that the incidence is in reality underdiagnosed and not that rare. finally, we discussed the higher rates of c.striatum infections in immunosuppressed patients.
We really hope that the modifications that we made will result in the acceptance of our paper for publication. thank you very much again for all your precious assistance.
Yours Sincerely,
The Authors
Round 2
Reviewer 3 Report
Comments and Suggestions for Authors
The authors have addressed the critique and have revised their manuscript adequately.